# “I’m Hooked on e-cycling, I Can Finally Be Active Again”: Perceptions of e-cycling as a Physical Activity Intervention during Breast Cancer Treatment

**DOI:** 10.3390/ijerph20065197

**Published:** 2023-03-15

**Authors:** Kirsty Mollie Way, Jessica Elizabeth Bourne, Miranda Elaine Glynis Armstrong

**Affiliations:** Centre for Exercise, Nutrition & Health Sciences, School for Policy Studies, University of Bristol, Bristol BS8 1TZ, UK

**Keywords:** electrically-assisted bicycles, e-bikes, breast cancer, qualitative, barriers, facilitators

## Abstract

Electrically-assisted bicycles (e-bikes) are a means through which to increase individual physical activity (PA) and overcome some commonly reported barriers to engaging in conventional cycling. Fatigue is a common side effect to breast cancer treatment, and the rate of PA engagement drops significantly following a breast cancer diagnosis. The aim of this qualitative study was to examine perceptions of e-cycling as a means of increasing PA in this population. Twenty-four participants (mean age = 57.88 (standard deviation 10.8), 100% female) who have had a breast cancer diagnosis, completed two semi-structured interviews via Zoom. One interview was conducted prior to an e-bike taster session and a second, after the session. Taster sessions were conducted by certified cycling instructors in the community. Interviews were conducted between December 2021 and May 2022. Data were transcribed verbatim and analyzed thematically using NVivo 12 software. An inductive and deductive approach to analysis was adopted. Five themes were generated: (1) Perceived role of e-bikes during treatment, (2) The relationship between e-bikes and fatigue, (3) Cancer-specific considerations, (4) Is e-cycling ‘enough’?, and (5) Optimizing the intervention. Negative perceptions of e-bikes noted before the taster session were altered following riding an e-bike. The multiple levels of assistance made cycling manageable and less impacted by fatigue, thereby enabling individuals to re-establish previous cycling habits. E-cycling may be a suitable option to increase PA behavior amongst individuals being treated for breast cancer, with the potential to overcome many of the barriers of conventional cycling. Enabling this population to trial an e-bike elicits positive physical and psychological responses that may help to promote future engagement.

## 1. Introduction

Cancer is one of the leading causes of mortality worldwide, totaling approximately 10 million deaths in 2020 [1,2]. Breast cancer is the most common cancer in the UK, accounting for 15% of all new cancer cases and approximately 55 million diagnoses, yearly [3]. There are a multitude of risk factors which contribute to the disease’s incidence, mortality, and survival rates, including: lifestyle factors, hereditary contributions, socioeconomic status, environment, and culture [4,5,6]. Despite some men developing the condition (1% of UK breast cancer cases) [3], breast cancer prevalence is greatest in women aged 50 and over, with ageing being the second greatest risk factor for the disease behind sex [7].

Cancer places great economic burden on society, with the cost of breast cancer chemotherapy alone mounting to over GBP 248 million, yearly [8]. Yet the financial costs of breast cancer can stretch further than direct treatment, with additional funding required for short- and long-term work absence, informal care provision, and carer emotional well-being [8]. Due to the high prevalence of breast cancer and associated costs, there is a need to identify effective lifestyle modifications for the tertiary prevention of breast cancer and associated mortality.

### 1.1. Breast Cancer and Physical Activity

Physical activity (PA) has been identified as a highly effective, non-pharmaceutical intervention known to complement other breast cancer treatments and modify recurrence rates [9]. The benefits of engaging in PA are obtainable even if the activity is delayed until 12 months post diagnosis [10]. PA guidelines for a breast cancer diagnosis do not differ from the conventional guidelines. As such, individuals should aim to partake in 150 min of moderate intensity activity per week, or 75 min of vigorous activity per week, with resistance training incorporated twice a week [11]. A prospective study of 1340 breast cancer patients found that individuals who met the PA guidelines both before and one year post diagnosis had a 41% reduced risk of breast cancer recurrence, and 49% reduced risk of mortality from breast cancer, when compared to physically inactive individuals; these associations were strengthened after 2 years (65% and 68%, respectively) [12].

In addition, in oncology, PA engagement is recognized to improve both psychological and physiological functions, including the potential to mitigate common side effects such as incidence of fatigue, impaired health-related quality of life (HRQL), mental health decline, and reductions in physical health [13].

It is important to note that an individual’s willingness and their physical and psychological responses to PA can vary depending on the type of treatment they are receiving. A cross-sectional study of 37 breast cancer patients revealed that patients were more likely to experience perceived difficulty engaging in PA if they had undergone more than three types of cancer treatment, in comparison to patients that experience no perceived difficulty (RR 2.14; 95% CI 1.07 to 4.27) [14].

### 1.2. Participation Rates and Barriers to Physical Activity

Despite the known benefits of PA and the intention to increase their PA participation once diagnosed, the number of women meeting national PA guidelines is low [15]. It is deemed challenging to find the motivation and capability to engage in PA following treatment, which is described as a debilitating process, thus limiting physical ability [16].

In 2012, a cohort study revealed that only 48.3% of breast cancer patients met the PA guidelines of 10 metabolic equivalent (MET) hours/week [17]. While these guidelines are outdated, they do suggest a decline in PA participation post diagnosis. A 10-year cohort study of 634 breast cancer patients identified that pre-diagnosis, 34.0% of women met the PA guidelines; this percentage stayed consistent at 24 months post enrollment (34.0%) but decreased to 21.4% after 10 years [18].

There are several perceived barriers to PA engagement during breast cancer treatment, including physical, psychosocial, environmental, and organizational factors [19]. Physically, women report impairments that arise due to surgery, such as shoulder problems, that limit PA engagement [19,20,21]. However, the most commonly reported barriers to PA engagement are lack of energy and fatigue, despite evidence to suggest that PA is beneficial for improving fatigue and boosting energy levels [19,22,23]. As such, it is important to identify different modalities of PA that are appealing to this population to promote engagement [24].

Smith-Turchyn and colleagues’ [25] qualitative research details how healthcare professionals vary the PA guidance they provide to patients depending on the treatment the patient is undergoing. Despite research promoting the benefits of PA during adjuvant chemotherapy [26], women tend to show an unwillingness to participate due to side effects such as nausea, fatigue, and emotional shock. In contrast, a five year longitudinal study found women undergoing chemotherapy gradually increased their participation in PA during the first 18 months of treatment, followed by a steady decline thereafter [27].

Emery and colleagues’ [27] research found that women are more likely to undertake PA if they had a lumpectomy, not a mastectomy. Therefore, it is important to note that physical implications resulting from surgery, such as restricted arm movement, can impact an individual’s perceived ability to take part in PA, as well as their self-efficacy and desire to be physically active [28]. Limited research has been conducted regarding when it is best to implement a PA intervention during breast cancer treatment.

### 1.3. The Emergence of E-Bikes

Electrically-assisted bicycles (e-bikes, also known as pedelecs) have been identified as a means through which to increase PA, providing at least moderate intensity PA [29]. Electrical assistance is administered only when the rider is pedaling via sensors and a motor [30]. With the additional electrical assistance, riders are motivated to cycle for longer periods of time and longer distances, thus increasing their overall PA engagement [31]. Specifically, survey results from over 10,000 individuals in seven European cities found that the average cycle duration was longer on an e-bike (35.0 min, 95% CI 31.7 to 38.3) compared to a conventional bike (25.9, 95% CI 25.4 to 26.5) [32]. In addition, individuals cycled further on an e-bike than on a conventional bike (9.4 km, 95% CI 8.6 to 10.2 and 4.8 km, 95% CI 4.7 to 4.9, respectively).

For older adults, e-cycling is reported as being preferable over conventional cycling due to the reduced physical exertion required on an e-bike, leading to reduced feelings of fatigue following a cycling journey [33]. Use of an e-bike has been reported to increase the user’s PA. A cross-sectional study of 340 Norwegian residents reported that PA increased by 353.9 min per week due to e-bike use [34].

Associations have been identified between e-bike use and physiological parameters. A longitudinal study by Dons and colleagues [35] found that body mass index (BMI) was −0.010 kg/m^2^ (95% CI −0.020 to −0.0002) lower per additional day of e-cycling per month. Additionally, maximal power output increased for both untrained men and women (192.19 ± 28.7 watts and 145.9 ± 24.8 watts, respectively) following six weeks of active commuting on an e-bike [31]. Furthermore, among sedentary women, e-cycling has been associated with greater levels of enjoyment compared to conventional cycling, whilst still eliciting sufficient levels of energy expenditure and power output [36]. As such, e-cycling is proposed to be a successful method to increase PA amongst sedentary women. Based on these findings, it is possible that e-cycling could be an appropriate means through which to introduce breast cancer patients to PA while undergoing treatment. At present however, no research has examined the role of e-cycling amongst breast cancer patients.

### 1.4. Aims and Research Questions

The aim of this research was to examine perceptions of e-cycling amongst individuals who have had a breast cancer diagnosis.

Based on this aim, the following research questions were developed:What are the perceived barriers and facilitators to e-bike usage during breast cancer treatment?When, during cancer treatment, is perceived to be optimum for introducing e-cycling?Does the implementation of an e-bike taster session elicit changes in perceptions of e-cycling during breast cancer treatment?

## 2. Materials and Methods

### 2.1. Study Design and Protocol

Due to the novel exploration of e-cycling during cancer treatment, a qualitative approach was identified as the most appropriate research method to identify initial perceptions and insights.

Two one-to-one semi-structured interviews were conducted with each participant; one before an e-bike taster session (referred to as Interview One) to gather the participants’ initial thoughts of e-cycling, and one post the e-bike taster session (referred to as Interview Two) to identify whether e-cycling opinions had changed. Semi-structured, one-to-one interviews were used due to their ability to keep the interview focused whilst still providing flexibility for the researcher to explore pertinent ideas that may be raised during the conversation [37]. Interviews took place virtually, without the presence of non-participants. Interviews were recorded using the Zoom record option and transcribed verbatim using the Zoom auto-transcription option. Further editing of the transcriptions was required to ensure accuracy in the data. Transcripts were not returned to the participants for comment or correction as they were anonymized.

Twenty-four interviews were conducted for Interview One. Following this, five participants were no longer contactable and did not complete the e-bike taster session or Interview Two. Four participants did not require a taster session due to already owning an e-bike. As such, fifteen participants completed the taster session and nineteen completed Interview Two.

### 2.2. Researcher Characteristics

All data collection was conducted by a female researcher, KMW (MSc). KMW was completing their MSc in Nutrition, Physical Activity and Public Health at the University of Bristol at the time of the study.

### 2.3. Participant Recruitment

#### 2.3.1. Inclusion and Exclusion Criteria

Participants were eligible to take part in the study if they have had a breast cancer diagnosis. Metastatic cancer is prominent amongst breast cancer patients, therefore individuals with secondary cancer were included [38]. Participants were excluded from the study if they did not consent to take part in an e-bike taster session, were unwilling to travel to Bristol for a taster session, or if they were unable to ride a conventional bicycle. For individuals who already owned an e-bike and were confident using it, it was not a requirement to attend the taster session.

#### 2.3.2. Sampling

A convenience sampling technique was used based on the aforementioned inclusion criteria. Participants were recruited via a Bristol-based charity, Penny Brohn UK. Advertisements were placed in two instalments of the monthly newsletters and emails were sent to a consenting mailing list. Additional social media adverts were promoted via Breast Cancer Now, on platforms including Facebook and Instagram. All advertisements contained the researcher’s contact details.

The recruitment reach through Penny Brohn UK was 281 via email mailing list and 4800 via monthly newsletter. From this, 34 individuals expressed initial interest in the study (approximately 0.7% of those reached), were sent the participant information sheet (int), and were presented with a link to an online demographic survey. The demographic survey collected data which allowed the researcher to screen individuals for eligibility; a consent form for the survey was also attached which participants were asked to sign and date electronically.

In total, 29 out of 34 who were sent the PIS completed the demographic survey. Of these 29, one was not eligible for the study due to not having a breast cancer diagnosis, and four dropped out due to personal reasons unrelated to the study. Data collection and analysis were conducted in parallel, therefore participant recruitment continued until saturation was met [39]. Data saturation was met after 20 pre-taster interviews, as enough data were collected to generate relevant themes and codes that were repeated across participants and it was assumed no additional useful insights would be provided with additional participants. However, four extra participants were interviewed to account for drop-out rates and their data was included in analysis (Table 1). Twenty-four participants completed Interview One and nineteen completed Interview Two.

### 2.4. Interview Questions

Separate interview guides were developed for use during Interview One and Two (Appendix A). The average duration of Interview One was 33 min, 45 s and 26 min, 55 s for Interview Two. No field notes were made during the interviews.

Interviews were conducted by the primary researcher, KMW. Informal, ice-breaker questions commenced the interviews followed by open-ended questions with probing questions included ad hoc, depending on the participant’s response, with the intention to deepen or expand on a point if necessary [40]. Pilot interviews were conducted to ensure coherence of the interview guides [41]. Interview questions were focused on understanding how cancer treatment impacted individuals’ PA, the potential barriers and facilitators to e-cycling during this time, and individuals’ opinion of when is best to introduce e-cycling during cancer treatment. In addition, questions focused on how perceptions of e-cycling changed following the trialing of an e-bike.

### 2.5. Taster Session

E-cycling is a recently popularized mode of PA, therefore it could not be assumed that participants had engaged in e-cycling previously [30]. To ensure responses to the interview questions were based on personal experience, participants were invited to take part in a free, one-hour e-cycling taster session following Interview One.

Taster sessions were conducted by qualified instructors at Life Cycle, a Bristol-based charity. A full safety briefing was delivered and safety equipment, such as high-visibility jackets and helmets, were provided on request. Participants were given an introductory ride on the e-bike around a Bristol park with a cycle path, which included both flat and hilly routes to ensure maximum exploration of the e-bike. Participants were given the autonomy to switch between levels of assistance and were provided with advice and support on how best to use the e-bike.

### 2.6. Ethical Approval

Ethical approval was obtained via the University of Bristol, School for Policy Studies Research Ethics Committee (Ethical Approval Number: EAN 055-21).

### 2.7. Data Analysis

Descriptive statistics were extracted from the demographic survey and presented in Table 1 as means and standard deviation (SD); analyses were performed using IMB SPSS Statistics, Version 28.

Interview recordings were held securely on the University of Bristol server until transcriptions were complete, after which they were deleted. All identifiable information was removed or anonymized during the transcription process and each participant was assigned an anonymous identification code.

Thematic analysis was used and guided by Braun and Clarke’s [42] six steps of analysis. Analysis was conducted concurrently with data collection (as described above) to determine saturation and iteratively generate codes and themes. NVivo 12 (OSR International Pty Ltd., New Delhi, India, v12, 2018) was used to carry out data analysis.

Transcripts were read repeatedly by KMW to ensure the researcher was familiar with the data. Following data familiarization, segments of text were highlighted, and initial codes were derived inductively and deductively (i.e., based on pre-specified research questions). KMW and a second coder (TJC) independently coded three transcripts. These three transcripts were selected by KMW to reflect diverse responses. The researchers met to discuss and refine the codes and a coding framework was developed. The remaining transcripts were coded by KMW, who revisited previously coded transcripts as required if new codes were identified. Codes were organized into categories based on their content through discussion between KMW and the second coder. From these categories sub-themes and higher order themes were generated.

To adhere to the quality criteria for all qualitative research, credibility, transferability, dependability, and confirmability were all considered [43], therefore a COREQ checklist was followed extensively when producing this report (Appendix A).

## 3. Results

Five overarching themes were generated from a combination of Interview One and Two: (1) Perceived role of e-bikes during treatment, (2) The relationship between e-bikes and fatigue, (3) Cancer-specific considerations, (4) Is e-cycling ‘enough’?, and (5) Optimizing the intervention (Figure 1).

### 3.1. Theme 1: Perceived Role of E-Bikes during Treatment

#### 3.1.1. Cycling Ability/Participation

Participants explained that e-bikes could play a significant role in overcoming barriers to cycling that are present following a diagnosis. For some participants, despite having a history of regular cycling, they were no longer able to participate because of their treatment due to factors such as reduced energy, weakness, fatigue, and lacking motivation:


*P.20: “I liked cycling, but it was hard work. Yeah. And, and now, now that I’m diagnosed and I’m on medication, I don’t think I could do it. I’m just exhausted after physical activity.”*
(Pre-taster, 96 months since diagnosis (SD), multiple cancer diagnoses, undergoing treatment)

Yet, the prospect of an e-bike was appealing, with the electric motor offering the required assistance to get participants back into their previous cycling habits:


*P.16: “I don’t use my push bike very much at all. I was reluctant to get on it I think for a long time because I hadn’t got the energy levels to do that. So yes, I think an e-bike definitely would help, knowing that you know I’ve got the power behind me.”*
(Post-taster, 15 months SD, undergoing treatment)

Participants discussed that an e-bike would facilitate an improvement in cycling performance, providing the ability to cycle further, faster, and with more confidence. Most prominently, the e-bike was perceived to overcome the inability to cycle up hills, due to strength reductions and deterioration in cardiovascular fitness that result from treatment:


*P.2: “It makes it more do-able, you’d be able to go further distances and hills wouldn’t put you off. So you didn’t, you wouldn’t have to be as fit to cycle with them.”*
(Post-taster, 49 months SD, undergoing treatment)

One participant compared up-hill e-cycling to feeling the same as cycling a conventional bike on a flat road, emphasizing that the electrical assistance allowed them to overcome their fear of hills:


*P.5: “Then we went up on a hill and of course she told me that I can use a higher program to help, then it was just like same like I was going on a straight, you know, it was like cheating almost, feeling that I’m cheating because that’s not the reality.”*
(Post-taster, 9 months SD, undergoing hormone treatment)

#### 3.1.2. Cycling with Friends and Partners

Cycling was often described as a social sport, with participants enjoying a leisurely ride with their families, friends, or community cycling groups. However, this was often diminished once treatment started, having a negative impact on the participants, both socially and psychologically:


*P.14: “My partner, and my daughters would go out riding bikes. But I, I wouldn’t go I couldn’t go because I, you know, I just hold them up, I couldn’t keep up with them.”*
(No-taster session as owned e-bike, 16 months SD, undergoing chemotherapy)

Yet, the e-bike was positively described as a way to overcome this barrier, due to the electrical assistance allowing for increased duration, thereby boosting their confidence to keep up with their peers and enhancing enjoyment:


*P.20: “There’s lots of people that are using e-bikes to like, so that they can carry on cycling with their partner because their partner is so much fitter and faster than them.”*
(Pre-taster, 96 months SD, multiple cancers, undergoing treatment)

#### 3.1.3. Positive Impact on Psychology

There were several physical benefits of e-cycling described by participants both pre- and post-taster session (Table 2), such as improving fitness, strengthening muscles, and getting the blood pumping to improve the dispersion of treatment drugs; yet the most prominent benefits were described to be psychological. Specifically, the act of getting outside and into the fresh air was enough to promote e-cycling amongst participants:


*P.6: “It’s just a wonderful feeling on an e-bike, the freedom the bike gives you, the wind in your hair and going through the countryside is just glorious, and just be able to continue to do that I think is really really important.”*
(No taster as owned e-bike, 32 months SD, undergoing hormone treatment)

Many participants also described the benefit e-cycling will have to their treatment response. In particular, the e-bike was commonly associated with alleviating the stress of both a diagnosis and when undergoing treatment:


*P.15: “I was very stressed to begin with, not quite knowing where it was all going. And, yeah, to me exercise is a way of releasing stress very definitely, and I think e-cycling is perfect in that way, even more than walking for me.”*
(Pre-taster, 17 months SD, undergoing hormone treatment)

For some, the taster session raised concerns that introducing e-cycling during treatment could lead to added stress, particularly amongst participants with limited cycling history and a fear of road cycling:


*P.5: “I guess, because it’s a kind of stress for me that I’m afraid of the traffic. And it was a stress for me to even just, you know, we mostly used the bike road but there were other bikers, there were there were other pedestrians.”*
(Post-taster, 9 months SD, undergoing hormone treatment)

Often, negative perceptions of e-bikes noted before the taster session were altered following a trial on the e-bike, with participants stating that cycling was “easier” than expected and a more “tolerable” mode of PA, thus making them more inclined to e-cycle in the future.

#### 3.1.4. Loss of Identity

Many participants reported a loss of identity or feeling like their cancer defines them. Yet, the prospect of an e-bike provided a sense of independence that they had been lacking since receiving their diagnosis:


*P.16: “Having the e-bike gives you that security but allows you to have the exercise as well, and and the independence, because otherwise you’re relying on someone else drive you or you have to try and park the car.” *
(Post-taster, 15 months SD, undergoing treatment)

For some, their diagnosis meant they were no longer able to drive. As such, the e-bike would provide a mode of travel that is quicker and more practical than walking:


*P.21: “I can’t drive so, I could use the e-bike for exercise and also if I ever wanted to sort of get to a shop or something and you know use it for that, that kind of thing.”*
(Post-taster, 56 months SD, undergoing chemotherapy)

### 3.2. Theme 2: The Relationship between E-Bikes and Fatigue

#### 3.2.1. Fatigue as a Barrier to Cycling

Table 2 highlights the key reported barriers and facilitators to e-cycling. The biggest barrier to conventional cycling during cancer treatment was said to be lacking energy and fatigue. This perception was therefore reflected onto e-cycling, whereby finding the energy to pedal was described as “unlikely” or “impossible” before trying an e-bike. This opinion largely diminished following the taster session:


*P.20: “I was surprised. I just didn’t feel exhausted. I think I was expecting it to take more out of me.”*
(Post-taster, 96 months SD, multiple cancer diagnoses, undergoing treatment)

The fear of not getting home due to tiredness once setting off on a conventional bike ride was enough to deter people from cycling completely. However, the use of an e-bike was assumed to address this problem:


*P.10: “If I cycled an hour there then that’s also an hour back. But with an e-bike, I feel like you just, you’ve that extra assistance so less worry about getting back, so then you just go a little bit further, I think. I’d be encouraged to go to places that you wouldn’t on a push bike.”*
(Pre-taster, 17 months SD, undergoing treatment)

Yet, anxieties were still raised post-taster session at the thought of the e-bike battery running low or stopping completely, thus leaving participants hesitant about cycling too far:


*P.8: “If you went on a route that is, like, I’m only choosing this route because I’ve got an electric bike so I know I can make it and then the bike breaks and then you’re like, oh what now.”*
(Post-taster, 17 months SD, undergoing hormone treatment)

#### 3.2.2. Conserving Energy

Although some participants expressed a desire to e-cycle when fatigued, others stated it would be detrimental to their health if they exerted themselves too much:


*P.15: “You have to be very careful not to overdo it, because it’s quite exciting to get on an e-bike, you know, and you think oh I’ll have a go, but it’s easy to do too much and suffer from it.”*
(Post-taster, 17 months SD, undergoing hormone treatment)

Emphasis was placed on the importance of conserving energy for treatments and therefore not participating in e-cycling on days of high fatigue:


*P.14: “If I lacked energy, I wouldn’t use an e-bike, I would be trying to conserve my energy for areas where I would need that energy.”*
(No-taster session as owned e-bike, 16 months SD, undergoing chemotherapy)

### 3.3. Theme 3: Cancer-Specific Considerations

#### 3.3.1. Physical Impairments

As a result of a lumpectomy or mastectomy, the restrictions in the arm and armpit were feared to limit the ability to hold the handlebars on the e-bike. However, the taster session appeared to eliminate this concern:


*P.9: “I thought maybe, you know, the slight weakness I have in my right arm because of the removal of the lymph node might have affected it, but that wasn’t a problem.”*
(Post-taster session, 8 months SD, finishing treatment)

For some, it was the strength in the arm that appeared most preventative:


*P.19: “I don’t think I had the strength in my hand or my arm to hold on and maintain a rigid arm. That was quite difficult.”*
(Post-taster, 92 months SD, multiple occurrences, undergoing treatment)

#### 3.3.2. E-Cycling Discomfort

E-cycling was sometimes described as a “bumpy ride”, with potholes or uneven road surfaces often inducing high, uncomfortable impact onto the bike and the rider. Some participants expressed concerns regarding this impact when undergoing treatment:


*P.17: “It wasn’t the exercise as much as the sort of the slight pummeling you get by going on the road on a small thing. So I guess that’s what would stop me using them at points when I was having chemo.”*
(Post-taster, 3 months SD, undergoing radiotherapy)

As a result, it was suggested that an individual’s stage of cancer should be considered when prescribing e-cycling as a mode of PA:


*P.20: “I think it’s something you’d, if you’re gonna recommend it you to people with stage 4 you’d have to take it on board, you’d have to find out more about their physical condition and assess if it’s suitable for them, I think.”*
(Post-taster, 96 months SD, multiple cancers, undergoing treatment)

Yet, participants recognized that the e-bike is no more prone to pummeling when compared to a conventional bicycle:


*P.18: “I tend to sort of like stand up on the pedals if I know there’s a bump coming or try and avoid it, or whatever. But it’s no more of an issue than it is with an ordinary bike.”*
(No taster as owned e-bike, 233 months SD, multiple cancers, undergoing hormone treatment)

### 3.4. Theme 4: Is E-Cycling ‘Enough’?

#### 3.4.1. Manageability

Following the taster session, a few participants reported that they expected the e-bike to be a strenuous form of PA that was potentially unmanageable, however, the taster session changed their perceptions:


*P.20: “The fact that we we went on this this bike ride we got back, and I just didn’t feel exhausted. I think I was expecting it to take it more out of me.”*
(Post-taster, 96 months SD, multiple cancers, undergoing treatment)

Others’ preconception was that e-cycling may not have provided any form of PA due to the assistance. However, following the taster session, e-cycling was perceived as being of sufficient intensity to provide a sufficient workout:


*P.19: “I think it was more than I expected because you still feel as though you’re cycling. You don’t feel as though everything’s doing the hard work for you. You feel that you’re having exercise, but just having that extra push when you needed it.”*
(Post-taster, 92 months SD, multiple occurrences, undergoing treatment)

Participants stated that the bike ride raised their heart rate sufficiently, worked their leg muscles, and induced muscle soreness the following day; all of which were claimed to be an indication of a good amount of physical exertion without the expense of fatigue:


*P.21: “But I was still puffed out when we got to the top, I had to have a little breather. You know I had the full support on I did still need a bit of a breather.”*
(Post-taster, 56 months SD, undergoing chemotherapy)

#### 3.4.2. Level of Assistance

Multiple assistance levels allowed participants to alter their exertion depending on how they were feeling, which was reassuring when experiencing fluctuations in energy and motivation during treatment:


*P.15: “So because I’m feeling fairly fit at the moment, I only had it on the eco level. So, yeah, even though it’s an e-bike, you can choose your level so that’s why it would be quite good because you could sort of tailor it to different people and how that person feels at the time”*
(Post-taster, 17 months SD, undergoing hormone treatment)

Some participants were willing to cycle with little or no assistance to ensure they made the most of the workout:


*P.7: “Once I finally got used to motor I turned it right down, because I wanted it to be part of my daily exercise.”*
(Post-taster, 97 months SD, undergoing hormone treatment)

However, the weight of the e-bike was frequently presented as a barrier to e-cycling (Table 2), and therefore the reason many participants would always cycle with at least some electrical assistance.

Some participants claimed the e-bike played a type of “psychological trick” on them and their willingness to cycle. Despite feeling tired and fatigued, participants suggested they would be more inclined to cycle an e-bike rather than a conventional bike, despite potentially exerting the same amount of energy:


*P.13: “It could have a psychological effect because you know you wouldn’t maybe go out on a normal bike, because of how you’re feeling yet you’d go out on an e-bike and end up actually doing the same amount of effort.”*
(Post-taster, 90 months SD, no current treatment)

### 3.5. Theme 5: Optimizing the Intervention

The opinion of whether e-cycling was possible during specific treatments varied greatly between participants, with only one common conclusion: it is solely down to the individual:


*P.1: “In general, whether you’d be able to do it with cancer treatments is individual because everyone is so different and everyone’s regime is different.”*
(Post-taster, 41 months SD, no current treatment)

#### 3.5.1. Diagnosis

Regardless of whether cycling throughout breast cancer treatment was seen as beneficial, most participants suggested not to promote an e-bike at the start of the cancer journey, following diagnosis:


*P.20: “It’s not when you’re first diagnosed, because it was just like you couldn’t even think something new.”*
(Post-taster, 96 months SD, multiple cancers, undergoing treatment)

However, P.7 suggested using the e-bike as soon as possible following diagnosis to set the patient up best for the treatment that follows:


*P.7: “The reasons they need it before is so that they can use it to improve their fitness, because the more fit you are before your surgery, the quicker recovery when you come out for chemotherapy.”*
(Post-taster, 97 months SD, undergoing hormone treatment)

#### 3.5.2. Surgery

Regarding surgery, it was popular to assume that the e-bike would not be used until the individual had recovered physically from the surgery. The recovery time appeared to depend on the type of surgery administered:


*P.9: “I guess it depends what surgery you have. If you’ve just had a lumpectomy, you probably could get onto a bike relatively soon.”*
(Pre-taster session, 8 months since diagnosis, finishing treatment)

Typically, post-surgery was claimed to be an ideal time to present an e-bike. In particular, the e-bike was described to act as a distraction away from the disease and the upcoming treatments (depending on the individual’s order of treatments):


*P.6: “It would have been the mindfulness of distracting me from all those dark thoughts and all that gloom and you feel like you haven’t got the future. So anything that takes you away from that.”*
(No taster as owned e-bike, 32 months SD, undergoing hormone treatment)

#### 3.5.3. Chemotherapy

E-cycling during chemotherapy presented the most controversial opinions. For some, despite chemotherapy being an exhausting time, they were confident they could continue using an e-bike depending on the week the chemotherapy was administered:


*P.13: “Week one, I wouldn’t be able to go for it. But then on week two and three, I was feeling perfectly well, or getting to the stage where I had enough energy to go for it.”*
(Post-taster, 90 months SD, no current treatment)

However, the prospect of e-cycling at any stage during chemotherapy seemed impossible for some:


*P.15: “I think chemo affects the brain too and they have that they say, you know that you have brain fog or kind of things and the dizziness and and all kinds of things which is probably not the very ideal time to go on a bike.”*
(Pre-taster, 17 months SD, undergoing hormone treatment)

#### 3.5.4. Radiotherapy

The main concern for e-cycling during radiotherapy was due to time commitments, rather than the treatment itself. The time necessary to attend appointments every day appeared to frequently result in fatigue, which then acts as a barrier to using the e-bike.


*P.17: “So, the tiredness wasn’t from the radiotherapy, it was more from the driving to the appointment, you know going to and from the appointment every day.”*
(Pre-taster, 3 months SD, undergoing radiotherapy)

The other barrier to e-cycling during radiotherapy was the skin burns:


*P.13: “You get radiotherapy burns. Would I have cycled with them? I don’t know, maybe.”*
(Pre-taster, 90 months SD, no current treatment)

#### 3.5.5. Recovery

Presenting the e-bike post treatment as a method to aid recovery seemed optimal amongst patients:


*P.20: “That sort of rehabilitation, that rebuilding yourself because after being poisoned and, you know, cut open and radiated, you know it’s a lot for your body to go through.”*
(Post-taster, 96 months SD, multiple cancers, undergoing treatment)

The e-bike was promoted as a method to return to “normal life”; a way of putting the cancer in the past.

## 4. Discussion

This is the first qualitative study to explore perceptions of e-cycling amongst individuals who have had a breast cancer diagnosis. With the aid of a 1-h taster session, this research highlighted the intention and desire for PA engagement during breast cancer treatment, with the e-bikes providing a promising method to overcome perceived barriers to physical exertion. Participants frequently referred to the “ease” of e-cycling, and its ability to re-introduce previous cycling habits that were disenabled because of their diagnosis and treatment. Reservations and considerations were reported regarding specific treatment side effects and the appropriate timing to introduce an e-cycling intervention. The findings of this research can be used by healthcare professionals when prescribing sustainable and manageable physical activity interventions alongside treatment. The emergence of themes and their alignment with current literature are considered below to determine the relevance and significance of e-cycling during breast cancer treatment.

### 4.1. The Impact of Taster Sessions

Table 2 shows an increase in the perceived barriers, facilitators, and benefits to e-cycling following the taster session. The importance of allowing participants to trial a piece of equipment to gather more detailed perceptions is demonstrated in research [44]. In the current study, it was common to have pre-conceived ideas about e-cycling, therefore the taster session was necessary to change attitudes and elicit more accurate perceptions of e-cycling in general, as well as its suitability during breast cancer treatment. Not only this, but trialing the equipment before use is important for user safety and confidence when initially using the e-bike. As highlighted in previous research, a pilot study which implemented a 12-week e-bike trial amongst breast cancer patients reported that, when asked if they were able to straightaway easily use the e-bike, participants answered 7 on a scale of 1 to 10 [45]. Although reasonably high, this demonstrates that e-bikes require practice, therefore the taster session is necessary to facilitate not only accurate perceptions of an e-bike, but also maximize the rider’s usability.

During pre-taster session interviews, it was evident that most participants recognized an e-bike’s image and usability, though were yet to consider its benefits for use during breast cancer treatment. Nevertheless, the prospect of e-cycling was desirable in the present study, with many participants claiming the taster session solidified their expectations that e-cycling would be an easier alternative to conventional cycling, allowing for increased distance and speed due to the electric motor. This aligns with previous research findings amongst e-bike owners in the Sacramento, California area, whereby the notion of increased distance inspired their purchase, with many participants disposing of their previously used conventional bicycles [46]. It is possible that e-cycling is appealing for individuals with breast cancer as it allows them to engage in a longer duration of exercise, something they deemed themselves incapable of following their diagnosis. A lost sense of identity is frequently reported in the present study, therefore managing to achieve what was previously a mediocre task could become fulfilling for individuals, allowing them to reconnect with their pre-diagnosis identities.

### 4.2. The Role of E-Bikes during Breast Cancer Treatment

#### 4.2.1. Physical Benefits

Participants expressed their perceived importance of PA during treatment through discussion of current exercise patterns. Following the e-cycling taster session, participants described the bodily fatigue and muscular soreness they experienced the next day, as well as the respiratory requirements of e-cycling. Previous research has confirmed the reduction in exertion from an e-bike, but also its suitability for individuals with physical limitations as a result [33]. Johnson and Rose [47] highlighted that 16.4% of elderly e-bike owners purchased their e-bike due to injury, illness, or disability, placing emphasis on their ability to now return to their cycling habits after previously being forced to stop. Further research also supports that e-cycling reduces stress on the body in comparison to conventional cycling and therefore increases its accessibility to many riders, thus providing a suitable alternative during breast cancer treatment [33,48,49].

Furthermore, frequent reference was made to the ability to cycle up a hill because of the electric motor, something which previously deterred individuals from getting on a bike. This reinforces previous research that recognized hills as a key barrier to conventional cycling and a key facilitator of e-cycling [50,51]. A 2002 survey of 600 UK e-bike users reported 37% of respondents stated the ease of use on hills as a main advantage of an e-bike over a conventional-bike [52]. It is possible that post-taster muscular soreness and difficulty with hill cycling reflects the fitness and deconditioning of the participants in this study, rather than the intensity of e-cycling itself. Many participants claimed they did not engage in cycling frequently prior to study engagement and would therefore not likely have the cycling-specific strength to cycle up hills with ease. In hilly terrains, it is important to consider the importance of hill assistance when expecting individuals undergoing breast cancer treatment to engage in cycling as a mode of PA.

#### 4.2.2. Psychological Benefits

Interestingly, participants discussed the desire to turn the electric motor off entirely when cycling on flat terrains. It was perceived that, during the taster sessions, participants felt the need to “push themselves, which consequently resulted in them working harder than anticipated due to the weight of the e-bike. This was described as “psychological trickery”, whereby the absence of the electric motor in fact promoted more intense PA and a greater sense of achievement. This is consistent with previous literature which explained that a perceived sense of achievement is a key facilitator to PA engagement during breast cancer treatment, in a study protocol where exercise instructors and continuous remarks of praise were used as a method to increase exercise self-efficacy [53]. With participants in the present study often expressing feelings of lacking control, it is possible they felt empowered when turning off the motor, demonstrating an ability to not only be in charge of the e-bike, but also their lives and the disease.

Enhancing the psychological benefits is important for improving long-term adherence to e-cycling and PA overall [54]. Research explains that intrinsic motivation is a key facilitator of PA participation, particularly if the activity is self-directed (i.e., more than 50% of the program is implemented without close supervision) [55], which an e-cycling intervention would be. As such, it is vital that individuals undergoing breast cancer treatment not only enjoy e-cycling, but find themselves motivated to take part. As described by the participants in this study, a heightened sense of achievement and “psychological trickery” is a promising reflection of sufficient intrinsic motivation.

### 4.3. Barriers and Facilitators to E-Cycling

#### 4.3.1. Fatigue

Most participants shared their relationship with fatigue during their treatment. Although fatigue was presented as a key barrier to conventional cycling, especially regarding hills and non-flat terrains, many suggested they would still engage in e-cycling when they would consider themselves too fatigued to engage in other, more strenuous types of PA. However, some days, such as during weeks of chemotherapy, e-cycling was deemed “impossible” or “unsuitable” due to the need to conserve energy. This supports previous research which identified a decrease in PA engagement when undergoing adjuvant chemotherapy, as well as significant decreases in energy expended through PA [56].

The greatest concern regarding fatigue was the fear of the e-bike battery running low and therefore cycling home with no assistance. Although advancements in technology mean that e-bike battery life is extending, this does not come without an increase in price [57]. Fears regarding battery life are common in research and, although not explicitly mentioned in the present study, there are concerns regarding remembering to charge the battery fully before use [46,58]. This is a consideration that must be accounted for when expecting breast cancer patients to cycle long distances as they already have an increased mental load due to their diagnosis [59].

#### 4.3.2. Accessibility

Despite listing extensive benefits and facilitators to e-cycling, participants were quick to identify limitations and barriers to this mode of PA. As e-bikes are a large financial investment, with prices increasing depending on the make and model [60], participants were overly concerned about safely storing and locking an e-bike. Many participants reported they would not leave it unattended in public areas, even with the use of a secure lock. Previous research reported similar findings, whereby the cost is enough to repel potential buyers or delay their purchase [61]. Notably, Bristol bicycle theft rates were at an all-time high in 2020; therefore, theft concern in the present study may be biased towards its Bristol-based demographic [62]. Furthermore, the socioeconomic status of users will determine the prevalence of the theft barrier; with socioeconomic status not measured in this study, it is difficult to determine its effect on study findings.

Prospectively, the initial investment for an e-bike would be eliminated if e-bikes were used within a future intervention amongst cancer patients, as the equipment costs would be covered by research funding. However, investment must be considered when looking to induce long-term e-cycling habits, when participation in research studies is complete. A survey study conducted in Norway found that higher perceived benefits and familiarity with e-bikes were positively related with the intention to buy an e-bike [63]. However, further research should be conducted within a UK sample, in addition to a demographic with a cancer diagnosis, to investigate whether individuals would be willing to invest in an e-bike during treatment. Particularly, there should be inclusion of a taster session, in line with Simsekoglu and Klöckner’s [63] findings that familiarity increases intentions to purchase an e-bike and will therefore promote long-term adherence.

#### 4.3.3. Social Interaction

A key reported facilitator and, consequently, benefit of e-cycling was the social opportunities that e-cycling provides. Frequently, individuals highlighted that an e-bike would allow them to cycle alongside peers and family, something they had not been capable of since their diagnosis. Rey-Barth and colleagues [45] found that group-based interventions involving e-bikes for breast cancer rehabilitation were beneficial for PA maintenance. The social aspect created environments for encouragement and increased enjoyment, with the homogenization of motor speed allowing for reduced extrinsic social comparisons. Interestingly, participants in the present study mentioned the appealing opportunity for group e-cycling, which aligns with Rey-Barth and colleagues’ [45] findings. It is possible that breast cancer patients experience social isolation following their diagnosis, which is particularly distressing when it results in exclusion from family and peer activities. The potential to re-engage in group cycling is likely ameliorating for participants, providing they eliminate any stigma or negative stereotypes that e-bikes often carry [64].

### 4.4. Cancer Treatment Considerations

Fears regarding physical limitations resulting from surgery, such as restricted use of the affected arm, are common in literature investigating the barriers to PA during breast cancer treatment. Participants in Sander and colleagues’ [65] study reported avoidance of resistance exercise, or modifications to workout routines, because of their joint pain and arm restrictions post surgery. Yet, this barrier was only expressed by a small number of participants in the present study, suggesting the demands of e-cycling may present less of a worry. When considering the optimal time throughout the cancer journey to introduce e-cycling, it was most frequently reported that immediately after diagnosis would be least convenient. Extensive research has supported the notion that a cancer diagnosis is a significant source of psychological stress and distress, with many patients struggling to cope with the initial shock of a diagnosis [66,67,68]. Specifically, research from Kang et al. [69] found that high cancer-specific stress was significantly correlated with high symptom perception amongst one hundred women with newly diagnosed breast cancer. When applying this finding to the present study, it could be argued that participants are less inclined to commence an e-cycling intervention due to higher perceived rates of symptoms, especially fatigue. Yet, research suggests that either continuing or commencing PA behavior immediately after diagnosis could be associated with up to 45% lower risk of death in comparison to women who were inactive both before and after diagnosis, (HR 0.55; 95% CI, 0.22 to 1.38) [70]. This suggests that the earlier the e-cycling intervention begins, the better. However, considerations regarding the participant’s preferences and willingness to adhere to exercise protocols if asked to engage when not ready must be recognized.

Uncertainty was displayed when deciding if e-cycling would be manageable and maintainable during other breast cancer treatments, in particular the cyclic nature of chemotherapy doses. Parallel to previous research, exercise adherence was reported lower during chemotherapy in comparison to radiotherapy (64% vs. 25%, *p* = 0.02) in a study where 68 breast cancer patients undertook regular supervized, moderate-intensity, aerobic, and resistance exercise [71]. All individuals respond to treatments differently which is likely to explain the inconsistencies in opinions of when to commence e-cycling. Additionally, intra-personal perceptions will likely fluctuate depending on the individual’s emotional and physical response to treatment. An optimal time to intervene is, therefore, difficult to distinguish.

### 4.5. Strengths and Limitations

To the author’s knowledge, this is the first qualitative study to explore the perceptions of e-cycling amongst individuals who have had a breast cancer diagnosis. The inclusion of e-bike taster sessions increased the validity of the participants’ perceptions, ensuring they were based on personal experiences rather than assumptions, and ensured that a change in perceptions pre- and post-taster session could be measured. The implementation of semi-structured interviews provided flexibility in the research questions, eliciting elaboration on points outside of the interview guide. Thematic analysis accurately followed Braun and Clarke’s [43] guidelines, and inductive reasoning was carried out to extensively analyze the data. Furthermore, the use of a second coder enhanced the credibility and confirmability of the research, and the quality of this review was maximized by following a COREQ checklist (Appendix A).

Nevertheless, the study’s limitations must be acknowledged. The expectation for participants to predict the feasibility and usability of an e-bike intervention during breast cancer treatment that they will not be taking part in could be deemed a difficult task, especially after receiving minimal information regarding the expected protocol. Furthermore, reporting and reflecting on previous feelings and emotions relies on accurate memory recall; importantly, the average time since diagnosis was more than a year (52.6 months; Table 1). Therefore, memories could be distorted, potentially impacting validity of research findings. The use of a single gatekeeper, Penny Brohn UK, to recruit participants had potential to implement bias towards a population that already engaged in healthy behaviors, thus limiting the study’s external validity. The requirement to travel to Bristol to attend the e-bike taster session likely limited the national reach of the study’s recruitment, therefore e-cycling perceptions could be biased towards the Bristol or Somerset landscape. As the participants’ residency location was not collected, it is difficult to measure the impact of this limitation on research findings. The lack of ethnic diversity in the study sample, as shown in Table 1, also limited the study’s generalizability to different cultures and may potentially contribute to ethnic disparities in healthcare research. Furthermore, the short timeframe of the study resulted in an absence of member-checking, a technique for exploring the credibility of qualitative results [72]. As a result, a lack of confirmability is acknowledged for the present study.

### 4.6. Implications for Practice and Research

The findings suggest that e-cycling may be a suitable form of PA for individuals undergoing breast cancer treatment. Specifically, e-cycling is reported to elicit both physical and psychological benefits and is consistently reported as a manageable form of PA. This suggests that the promotion of e-cycling during cancer treatment is appropriate. Specifically, with the right funding and provision, e-cycling could be offered by doctors, oncologists, and nurses alongside traditional treatments (such as chemotherapy and radiotherapy) to reduce or eliminate negative treatment side effects (such as reduced quality of life, mental health decline, and reductions in physical health) [13] and recurrence rates [9]. The reported psychological boost that e-cycling elicits could help patients to process the negative emotional state that accompanies treatment, thus contributing to a better quality of life [73].

Future researchers can be confident that breast cancer patients support the prospect of e-cycling during treatment and would be keen to partake if the opportunity was presented to them when undergoing treatment. Researchers must recognize that introducing an e-bike immediately after a cancer diagnosis could be detrimental, yet an optimal time to intervene is highly dependent on the individual. Participants in this study valued the taster session to gain a better understanding of the e-bike usability and practicality. Therefore, the implementation of an e-bike taster session will be essential when looking to recruit participants for a future e-bike intervention. Barriers to e-cycling predominantly surrounded initial investment, therefore it is imperative to ensure external funding is secured to cover costs of the e-bike if recruiting for a future intervention. Additionally, knowledge of how to access an e-bike once study involvement has concluded would contribute to long-term e-cycling engagement amongst cancer patients.

To address the limitations of the present study, future research should aim to gather perceptions from a more ethnically diverse population to reduce disparities in healthcare research. Additional further research should qualitatively assess the motivations, facilitators, and barriers behind buying an e-bike during a cancer diagnosis. Understanding these factors may assist with prolonging e-cycling engagement throughout cancer treatment and beyond.

## 5. Conclusions

This qualitative study provides novel insights to the perceived usefulness and practicality of e-bikes during breast cancer treatment. It is presented that individuals who have had a breast cancer diagnosis recognize the importance of engaging in PA during treatment and believe e-cycling is a suitable mode to overcome common barriers to conventional cycling, such as fatigue and physical exertion. There are both physical benefits of e-cycling, such as its suitability for individuals with a cancer diagnosis, and psychological benefits, such as fresh air and the influence to work harder than anticipated. However, worries regarding battery life and theft were prominent, and the initial financial investment deterred some participants from potentially purchasing an e-bike. An optimal time to introduce an e-bike intervention was difficult to discern for participants, as cancer treatment is explained to be highly dependent on the individual. Overall, e-cycling was deemed an appropriate mode of PA to engage in during breast cancer treatment. It suggests promise for increasing PA behaviors in this population, with the potential to overcome many barriers posed by conventional cycling. The positive physical and psychological responses associated with providing e-bike taster sessions in this population may help to promote future engagement.

## Figures and Tables

**Figure 1 ijerph-20-05197-f001:**
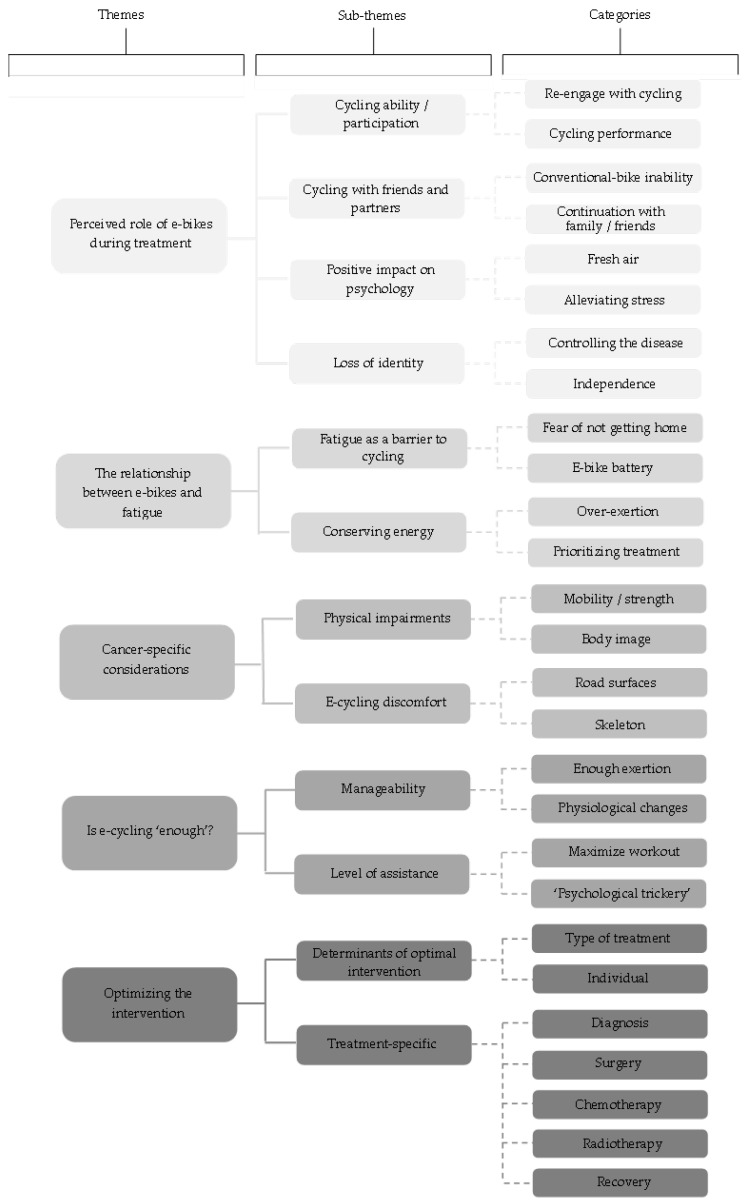
A hierarchical model of the themes generated for perceptions of e-cycling during breast cancer treatment.

**Table 1 ijerph-20-05197-t001:** Demographic characteristics.

Variable	N (%)
Age, years, mean (SD)	57.88 (10.8)
Sex	
Female	24 (100)
Male	0 (0.0)
Ethnicity	
White	22 (91.6)
Mixed	2 (8.3)
Asian/Asian British	0 (0.0)
Black/African/Caribbean/Black British	0 (0.0)
Other	0 (0.0)
Education	
Less than secondary school	0 (0.0)
Secondary school or equivalent	8 (33.3)
Bachelor’s degree	9 (37.5)
Master’s degree	5 (20.8)
Doctorate	1 (4.2)
Other	1 (4.2)
Employment status	
Full-time (35 h or more per week)	4 (16.7)
Part-time	6 (25.0)
Self-employed	2 (8.3)
Unemployed	2 (8.3)
Retired	8 (33.4)
Sick leave	2 (8.3)
Student	0 (0.0)
Marital status	
Married	17 (70.9)
Divorced	2 (8.3)
Widowed	2 (8.3)
Unmarried	3 (12.5)
Current treatment status	
Ongoing	20 (83.3)
Ended	4 (16.7)
Months since diagnosis, mean (SD)	40.4 (35.5)
Treatment type (combined across treatment window)	
Surgery (mastectomy/lumpectomy)	21 (87.5)
Chemotherapy	14 (58.3)
Radiotherapy	15 (62.5)
Hormone therapy	16 (66.7)
Other treatment	3 (12.5)
E-bike ownership	
Yes	4 (16.7)
No	20 (83.3)

Abbreviations: SD = standard deviation.

**Table 2 ijerph-20-05197-t002:** Perceived barriers, facilitators, and benefits of e-cycling during cancer treatment both pre- and post-taster session.

	Pre-Taster Session	Post-Taster Session (Additional)
Barriers	Appearance Battery life Energy/fatigue Expensive to buy Difficult to store safely Weight-heavy to pedal and maneuver	E-bike maintenance Theft Difficult to transport Remembering to charge Impact from road
Facilitators	Weather Someone to cycle with Cycling history Fitness	Cycle paths Knowledge of cycle routes Sufficient training Purposeful e-cycling groups Cycling ability
Benefits	Enhances mood Assistance makes cycling easier Environmentally friendly Independence Able to keep up with partner Prolong cycling duration Tackle hills Less fitness required	Enjoyment (more than conventional-bike) Replace car when commuting Holds space on road Safer than conventional-bike Conserve energy compared to conventional-bike Time savings

## Data Availability

The data that support the findings of this study are available on request from the corresponding author, MEGA, with the following restrictions: The data are not publicly available, and participants have restricted permission for use to those planning e-bike intervention studies in cancer patients.

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
