# Peer review of "“I’m Hooked on e-cycling, I Can Finally Be Active Again”: Perceptions of e-cycling as a Physical Activity Intervention during Breast Cancer Treatment"

_ijerph, 2023, doi:10.3390/ijerph20065197_

Round 1

Reviewer 1 Report

The authors claimed to assess the use of e-cycling in breast cancer patients on the perception of active stay.

The argument is interesting and new; the aims were clearly described; methods, statistics  are appropriated; conclusion meets the results only in part.

Major remarks:

-figure 1 was missed;

-Title must be changed i.e. by emphasizing the perception of e-cycling in brest cancer survivors instead of to assess the involvement in e-cycling;

-in the 4.6 paragraph lines 767-769 the authors must in deep discuss on the  improvement of physical fitness parameters related to the health in e-cycle users, considering the heart and muscular damage elicited by cancer treatment.   

Author Response

Thank you for the comments on our manuscript. We have addressed the comments, as detailed below and feel that the manuscript has been improved as a result. All changes to the manuscript are highlighted in the revised manuscript. 

The authors claimed to assess the use of e-cycling in breast cancer patients on the perception of active stay. The argument is interesting and new; the aims were clearly described; methods, statistics are appropriated; conclusion meets the results only in part.

Major remarks:

Figure 1 was missed

Thank you for pointing this out. We had mistakenly labelled Figure 1 as Figure 2. We have now corrected this on line 314.

Title must be changed i.e. by emphasizing the perception of e-cycling in breast cancer survivors instead of to assess the involvement in e-cycling;

Thank you for your comment. We have now changed the title to ‘ “I'm hooked on e-cycling, I can finally be active again”: Perceptions of e-cycling as a physical activity intervention during breast cancer treatment.’ and hope that this is now acceptable.  

in the 4.6 paragraph lines 767-769 the authors must in deep discuss on the  improvement of physical fitness parameters related to the health in e-cycle users, considering the heart and muscular damage elicited by cancer treatment.

This comment is unclear as the lines quoted do not contain materials relevant to your comment. Maybe you are referring to the concept of Cancer Cachexia? We note that recent research (Mallard J, Hucteau E, Hureau TJ and Pagano AF (2021) Skeletal Muscle Deconditioning in Breast Cancer Patients Undergoing Chemotherapy: Current Knowledge and Insights From Other Cancers. Front. Cell Dev. Biol. 9:719643) has concluded that there is still insufficient evidence around the mechanisms in breast cancer-mediated muscle deconditioning which would be needed before making specific exercise or physical activity recommendations. As studies are yet to be conducted on specific physiological changes elicited by e-bike use in those diagnosed with breast cancer, it would be premature to make any in depth comments on this in our paper. Hence, we have referred to this more generally under the umbrella of ‘physical health’ in line 797.

Reviewer 2 Report

This manuscript represents the study regarding the perceptions of e-cycling following a breast cancer diagnosis. These are my comments and suggestions:

Abstract:

Please, better formulate the goal and design of your research.

Introduction:

Introduction includes adequate theoretical background and references. However, please shorten the part regarding the aim of the study.

Methods:

Methods used are adequate and used according the guidelines. I would just suggest to better explain how did you conclude that your sample size is adequate. Also, give more details regarding the taster session of e-cycling.

Results:

Nicely presented.

Discussion:

Briefly explain/highlight why your study is important and what are the clinical and practical implications (beginning of the Discussion). For example, "this is the first qualitative study"...

The rest of the Discussion is excellent.

Author Response

Thank you for the comments on our manuscript. We have addressed the comments, as detailed below and feel that the manuscript has been improved as a result. All changes to the manuscript are highlighted in the revised manuscript. 

This manuscript represents the study regarding the perceptions of e-cycling following a breast cancer diagnosis. These are my comments and suggestions:

Abstract:

Please, better formulate the goal and design of your research.

We have now amended the abstract to clearly state the qualitative design of this study, as well as the research aim: to examine perceptions of e-cycling as a means of increasing physical activity amongst individuals with a breast cancer diagnosis. Lines 13-14

Introduction:

Introduction includes adequate theoretical background and references. However, please shorten the part regarding the aim of the study.

Thank you for your comment. We have now shortened the section to remove repetition and it is now clearer for the reader. Specifically, the sentence regarding when during treatment would using an e-bike be most beneficial has been removed. The statement remains as one of the three research questions.

Methods:

Methods used are adequate and used according the guidelines. I would just suggest to better explain how did you conclude that your sample size is adequate. Also, give more details regarding the taster session of e-cycling.

A definition of data saturation has been provided to clarify how saturation was met, and therefore how we concluded the sample size was adequate (Lines 194 to 196). A more detailed overview of the contents of the e-bike taster session and the location of the session has been provided, Lines 222 to 226.

Results:

Nicely presented.

Thank you

Discussion:

Briefly explain/highlight why your study is important and what are the clinical and practical implications (beginning of the Discussion). For example, "this is the first qualitative study"...

Thank you for your comment. We have amended the discussion accordingly to reflect the clinical and practice significance of the study (Line 581, lines 589 to 591). It has now been stated that this is the first qualitative study to explore perceptions of e-cycling amongst this demographic. Clinical implications have been included, such as the suggestion that healthcare professionals can use this information to prescribe more sustainable and manageable physical activity interventions.

The rest of the Discussion is excellent

Thank you

Round 2

Reviewer 1 Report

The authors modified the paper accordingly to the reviewer observation.